# Development of Stiffness Measurement Program Using Color Mapping in Shear Wave Elastography

**DOI:** 10.3390/diagnostics10060362

**Published:** 2020-06-01

**Authors:** Haneul Lee, Kyuseok Kim, Youngjin Lee

**Affiliations:** 1Department of Physical Therapy, Gachon University, 191, Hambakmoero, Yeonsu-gu, Incheon 21936, Korea; leehaneul84@gachon.ac.kr; 2Department of Radiation Convergence Engineering, Yonsei University, 1, Yonseidae-gil, Wonju-si 26493, Gangwon-do, Korea; 3Department of Radiological Science, Gachon University, 191, Hambakmoero, Yeonsu-gu, Incheon 21936, Korea

**Keywords:** ultrasound, shear wave elastography, stiffness measurement program, color mapping program

## Abstract

Shear wave elastography with ultrasound is a noninvasive method used for measuring stiffness in the human body. Shear wave elastography can be used for accurately and quantitatively measuring stiffness. However, its disadvantage is that the stiffness value can vary significantly because the region of interest (ROI) setting depends on the diagnostic operator. In this study, a stiffness measurement program using color mapping in shear wave elastography was developed to address the above-mentioned disadvantage. Color map and color ratios were obtained and evaluated for major lower limb muscles (i.e., biceps femoris, medial gastrocnemius, rectus femoris, and tibialis anterior) at active voluntary contraction. According to the result, when the developed program was used, a small standard deviation compared to the conventional stiffness measurement method, such as kilopascal or meter per second unit using ROIs, was measured in all cases. In conclusion, our results demonstrate that the stiffness measurement method using our program is expected to improve reliability in shear wave elastography ultrasound imaging.

## 1. Introduction

Computed tomography (CT) based on an X-ray source and magnetic resonance imaging (MRI) using radiofrequency pulses are widely used in the field of diagnostic imaging [1,2]. However, CT and MRI present problems of high exposure dose and high cost, respectively. Since the 1960s, ultrasound, which uses high-frequency soundwaves and can transmit via various mediums, has become one of the major scanning methods in the field of diagnostic imaging [3,4,5,6]. Currently, ultrasound is widely used to diagnose musculoskeletal pain in a muscle or tendon, particularly strain injuries, which involve tearing of muscles or tendons that can cause high or low elasticity and various diseases in the body [7]. Among the ultrasound imaging methods used for the diagnosis of musculoskeletal pain, sonographic elastography, which can be used to measure the elasticity of tissues numerically and visually, is the most widely used [8,9,10,11,12].

Palpation is one of the most widely used methods for evaluating diagnostic research. It involves a physical examination of the body to assess tissue stiffness, such as for muscles, tendons, lymph nodes, breast mass, and liver fibrosis [13,14]. However, the reliability of this technique is a major concern for the identification of a patient’s disease [15]. To address this concern, sonographic elastography using ultrasound was developed. Sonographic elastography is a promising imaging method and has applications in the field of oncology [16,17,18]. It obtains images based on tissue hardness information using a conventional ultrasound device with a software program [16]. Among the sonographic elastography imaging methods, shear wave elastography uses perpendicular stress force (shear wave propagates laterally in the object) and is based on the principle of imaging for elastic deformation in a medium. The propagation velocity of a shear wave is based on the stiffness of the tissue. The elastic modulus (Young’s modulus) of the tissue is calculated and measured in kilopascal (kPa) or meters/second (m/s) from the measured velocity. Malignant tumor tissues have higher measured values than benign tissues for the evaluation parameters. In general, an image obtained through shear wave elastography displays red color for a solid tissue (higher stiffness) and blue color for a soft tissue (lower stiffness). In addition, this elastography technique is widely used for accurate stiffness measurements of muscles or tendon tissues [12]. Figure 1 shows an example of an image of a hamstring obtained using shear wave elastography.

To perform quantitative evaluation from shear wave elastography images including muscle or tendon images, region of interest (ROI) setting is essential. However, the quantitative evaluation value with respect to the ROI settings vary significantly depending on the ultrasound operator. Thus, to develop a software program that can extract the overall color ratio of an image, a relatively high objectivity result can be obtained for shear wave elastography. In this study, we developed a program to measure stiffness using color mapping in shear wave elastography as well as to evaluate the color ratio based on the stiffness of major lower limb muscles.

## 2. Materials and Methods

### 2.1. Used Shear Wave Elastography Ultrasound 

Ultrasound images and shear wave elastography images were captured using an Acuson S-3000 ultrasound device (Siemens Healthcare Co., Ltd., Erlangen, Germany) with a linear probe at a high frequency (9.4 MHz). The linear probe consisted of a high-density element array with a fine pitch using 576 number of elements and was positioned in the middle region of each muscle, parallel to the fascia plane. The stiffness measurements were performed by a clinical sonographer with more than 8 years of experience in diagnostic radiology.

### 2.2. Procedure 

The study procedure was approved by the Gachon University Institutional Review Board. All participants signed an informed consent prior to participating in this study. The researcher marked measuring points on each muscle with measuring tape from the anatomical landmarks. Muscle stiffness was measured in at active voluntary contraction. All participants were instructed on how to contract their muscles before measurement. Four muscles of lower limbs were used for the stiffness measurements using shear wave elastography: biceps femoris (BF), medial gastrocnemius (MG), rectus femoris (RF), and tibialis anterior (TA). The positions of muscles contraction were explained in a previous study [19]. 

### 2.3. Used Shear Wave Elastography Ultrasound 

Ultrasound images and shear wave elastography images were captured using an Acuson S-3000 ultrasound device with a linear probe at a high frequency (9.4 MHz). The linear probe consisted of a high-density element array with a fine pitch using 576 number of elements and was positioned in the middle region of each muscle, parallel to the fascia plane. The stiffness measurements were performed by a clinical sonographer with more than 8 years of experience in diagnostic radiology.

The manufacturers of the ultrasound device provided a quality factor (Q-factor) for the stiffness measurements [20]. This factor indicates the accuracy of shear wave speed measurement and provides real-time performance to assist the operator [21]. In all the acquired images, we used Q-factors >80 to minimize the motion artifact.

### 2.4. Stiffness Measurement Program Using Color Mapping

We proposed an area distribution that represents the relative superiority of red, green, and blue images, and then calculated each color ratio, which can be expressed as:(1)$$I\left(x,y\right)=\{\begin{array}{c} I_{R}\left(x,y\right)\\ I_{G}\left(x,y\right)\\ I_{B}\left(x,y\right) \end{array}\;0\leq I_{R}\left(x,y\right),I_{G}\left(x,y\right),I_{B}\left(x,y\right)\leq 255,$$
(2)$$Map_{R}\left(x,y\right)=I_{R}\left(x,y\right)-I_{G}\left(x,y\right)\geq T\;and\;I_{R}\left(x,y\right)-I_{B}\left(x,y\right)\geq T,\;Map_{G}\left(x,y\right)=I_{G}\left(x,y\right)-I_{R}\left(x,y\right)\geq T\;and\;I_{G}\left(x,y\right)-I_{B}\left(x,y\right)\geq T,\;Map_{B}\left(x,y\right)=I_{B}\left(x,y\right)-I_{R}\left(x,y\right)\geq T\;and\;I_{B}\left(x,y\right)-I_{B}\left(x,y\right)\geq T,$$
(3)$$J_{R}\left(x,y\right)=I_{R}\left(x,y\right)\;\circ \;Map_{R}\left(x,y\right),\;J_{G}\left(x,y\right)=I_{G}\left(x,y\right)\;\circ \;Map_{G}\left(x,y\right),\;J_{B}\left(x,y\right)=I_{B}\left(x,y\right)\;\circ \;Map_{B}\left(x,y\right),$$
(4)$$Ratio_{R}=\overset{max}{\underset{x=1}{\sum }}\overset{max}{\underset{y=1}{\sum }}\left(J_{R}\left(x,y\right)/\left(J_{R}\left(x,y\right)+J_{G}\left(x,y\right)+J_{B}\left(x,y\right)\right)\right),\;Ratio_{G}=\overset{max}{\underset{x=1}{\sum }}\overset{max}{\underset{y=1}{\sum }}\left(J_{G}\left(x,y\right)/\left(J_{R}\left(x,y\right)+J_{G}\left(x,y\right)+J_{B}\left(x,y\right)\right)\right),\;Ratio_{B}=\overset{max}{\underset{x=1}{\sum }}\overset{max}{\underset{y=1}{\sum }}\left(J_{B}\left(x,y\right)/\left(J_{R}\left(x,y\right)+J_{G}\left(x,y\right)+J_{B}\left(x,y\right)\right)\right),$$
where I(x,y) is the ultrasound image at coordinates *x*,*y*, which is composed of $I_{R}\left(x,y\right)$, $I_{G}\left(x,y\right),$ and $I_{B}\left(x,y\right)$; $Map_{color}\left(x,y\right)$ is the index map based on the threshold value *T* (we used *T* = 30, heuristically); ∘ is an element-wise multiplication operator; and $Ratio_{color}$ is a scalar value.

Figure 2 shows a simplified flowchart of the proposed stiffness measurement process. An elastography ultrasound image is acquired from the sonography system (①), and $Map_{color}\left(x,y\right)$ is calculated using $I_{R}\left(x,y\right)$, $I_{G}\left(x,y\right),$ and $I_{B}\left(x,y\right)$ according to *T* (②). Finally, the color ratio value is obtained and an accurate stiffness measurement (③) is performed.

Based on the above descriptions, we implemented the proposed algorithm using the MATLAB™ v8.3 programming language. To process this software in the proposed program language, we performed the following the data process: First, we acquired the elastography data in DICOM format and then it is interlocked with MATLAB software to acquire 3-dimensional data (i.e., width, height, and depth (color channel)) without DICOM header information. Finally, each color result was derived by applying the proposed algorithm in the image domain.

## 3. Results and Discussion

Shear wave elastography allows the quantitative evaluation of tissue stiffness in units of kPa or m/s using the ROI and perform a musculoskeletal pain examination. In general, the ROI setting in the shear wave elastography image sets the number of target muscle or tendon areas, and then derives an average quantitative value. However, this elastography has not been used practically because the measured stiffness is different for each operator. Thus, the purpose of this study was to develop a more quantitative and objective stiffness measurement program using color mapping in shear wave elastography.

Figure 3 shows the stiffness measurement data for the contracted BF, MG, RF, and TA using shear wave elastography. Each stiffness value is expressed as an average of five ROIs. In kPa, the stiffness values were a mean of 100.60 ± 17.48 (standard deviation), 116.04 ± 26.53, 71.77 ± 14.20, and 155.52 ± 22.57 for the contracted BF, MG, RF, and TA, respectively. In addition, the stiffness values in m/s were 5.96 ± 0.98, 6.60 ± 1.54, 4.83 ± 0.89, and 7.82 ± 1.27 for the contracted BF, MG, RF, and TA, respectively. As a result, the conventional stiffness measurement method confirmed that the standard deviation changes significantly according to the ROI settings or operator.

Figure 4 shows the resulting images using the developed stiffness measurement program with color mapping based on shear wave elastography for the major lower limb muscles in contraction position. The developed program can acquire four types of images, including the original shear wave elastography image. Red, green, and blue color maps were extracted to indicate the stiffness of each tissue. A visual assessment of the images indicated that the higher the stiffness, the wider the red map (TA > MG > BF > RF). Conversely, we confirmed that the distribution of the blue map widens in tissues with low stiffness (RF > MG > BF > TA).

Based on the color extracted images, we evaluated the quantitative data for each color ratio with the major lower limb muscles (Figure 5 and Table 1).

According to the quantitative results, the red color ratio was a mean of 0.028 (approximately 2.8%) ± 0.0023 (standard deviation), 0.139 ± 0.0128, 0.018 ± 0.0015, and 0.347 ± 0.0161 for the contracted BF, MG, RF, and TA, respectively. The evaluated red color ratio increased in ascending order of RF, BF, MG, and TA. In addition, the blue color ratio had a mean of 0.423 ± 0.0265 (standard deviation), 0.552 ± 0.0368, 0.583 ± 0.0105, and 0.002 ± 0.0001 for the contracted BF, MG, RF, and TA, respectively. The evaluated blue color ratio increased in ascending order of TA, BF, MG, and RF. A comparison of the quantitative evaluation and visual assessment confirmed that the same tendency was observed. However, the green extraction results showed no tendency for tissue stiffness. The evaluated green color ratio increased in the order of MG, RF, BF, and TA.

The purpose of developing the stiffness measurement program using color mapping was to overcome the disadvantages of the conventional evaluating method using ROIs. As previously mentioned, the conventional method is not used practically and has a major drawback in that the deviation of the result becomes large because the ROI setting depends on the operator. The ratios of the differences between the mean and the standard deviations were 15.65% and 15.63% for kPa and m/s, respectively, with the conventional evaluation method. In addition, the ratios of the differences between the mean and the standard deviations were 6.97%, 5.18%, and 5.20% in the red, green, and blue maps, respectively, with our developed program. Based on the results, we confirmed that the reliability of our developed program is better than that of the conventional method for evaluating stiffness using shear wave elastography.

In this study, we developed a color mapping program to obtain a more accurate stiffness value than the conventional method in shear wave elastography. However, the limitation of this study was the stiffness range setting for shear wave elastography in major muscle scans. Kunwar et al. suggested an elastography scoring system for cervical lymph nodes and breast lesions [22]. In that study, a scoring system was developed in four stages from soft tissues to tissues with higher stiffness and was used in clinical practice. Based on the developed program, we expect that further research on the range of the stiffness interpretation of the obtained values will improve the diagnostic accuracy in the field of musculoskeletal disease diagnosis. To improve the diagnostic efficiency in ultrasound imaging, accurate analysis of the speckle pattern and near field clutter is crucial [23,24,25,26]. Speckle pattern and near field cluster are both very heterogeneous and it is expected that the method using a color mapping program will have superior characteristics compared to the method of extraction by adjustment to the conventional gray scale range.

## 4. Conclusions

We objectively evaluated the stiffness of major lower limb muscles using a color mapping software program with shear wave elastography imaging. Our results demonstrated that our developed program can improve the reliability and accuracy of quantitative evaluation of stiffness in shear wave elastography. Based on the results of this study, it is expected to be applicable to accurate measurement of stiffness of various muscles and tendons as well as major lower limb muscles.

## Figures and Tables

**Figure 1 diagnostics-10-00362-f001:**
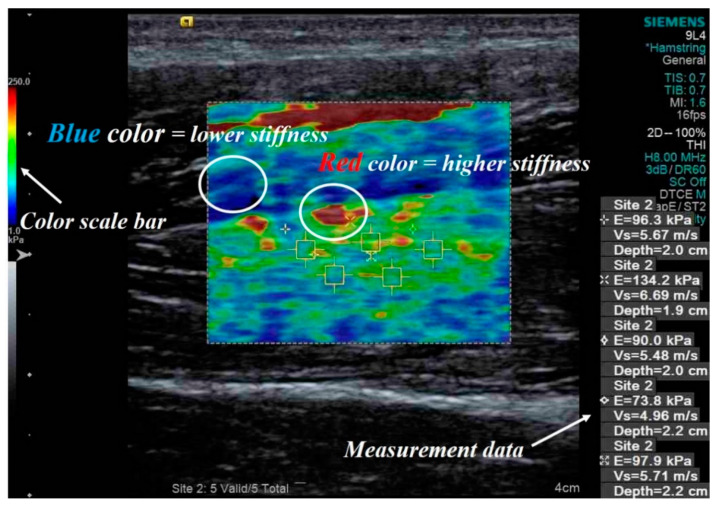
Example of a shear wave elastography image including quantitative measurement data expressed in kPa and m/s for a hamstring. The yellow box indicates the region of interest observed in the shear wave elastography image, and the white circles indicate the intensity of the stiffness expressed in blue and red colors.

**Figure 2 diagnostics-10-00362-f002:**
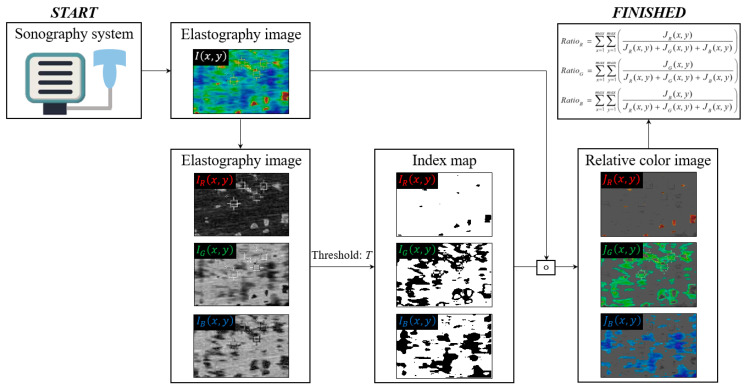
Simplified illustration of proposed stiffness measurement process using a shear wave elastography image.

**Figure 3 diagnostics-10-00362-f003:**
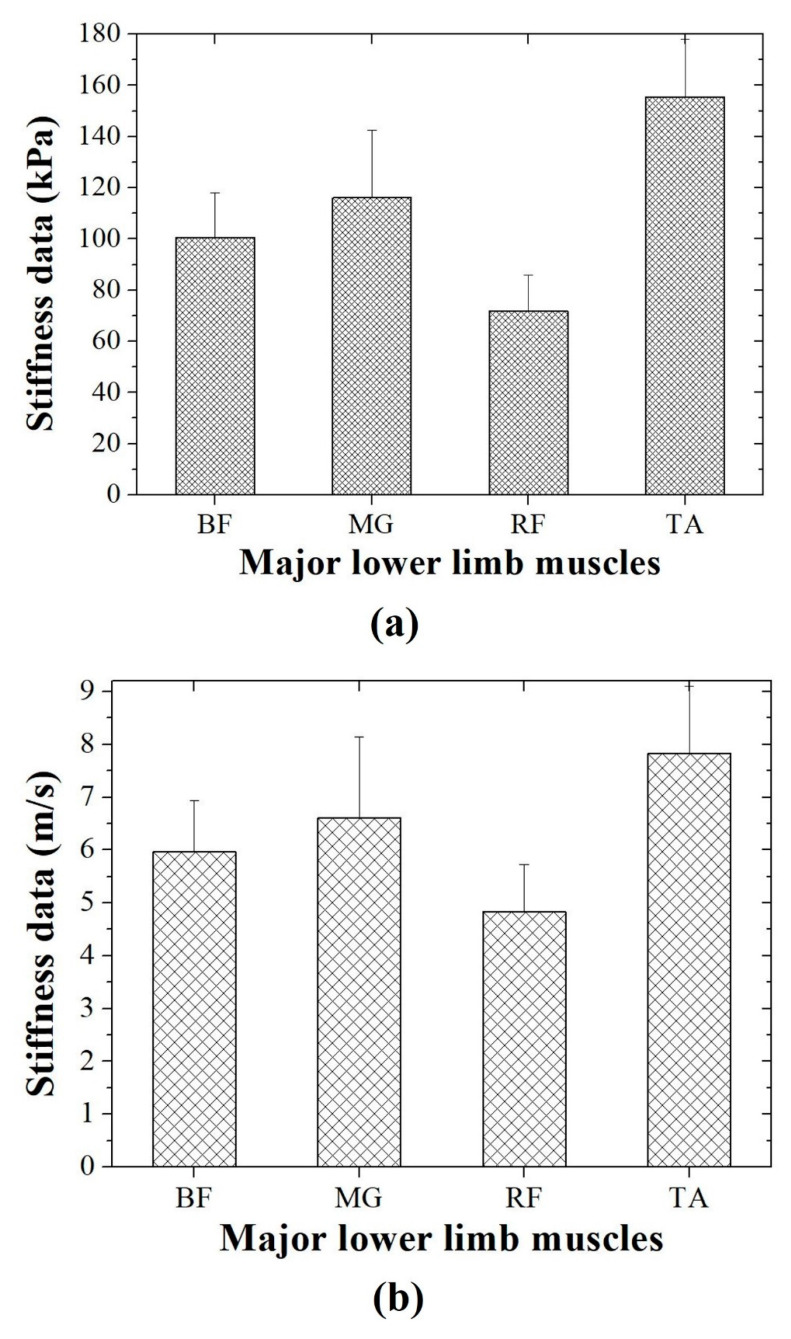
Acquired stiffness data in (**a**) kPa and (**b**) m/s in shear wave elastography for major lower limb muscles (BF: biceps femoris, MG: medial gastrocnemius, RF: rectus femoris, TA: tibialis anterior).

**Figure 4 diagnostics-10-00362-f004:**
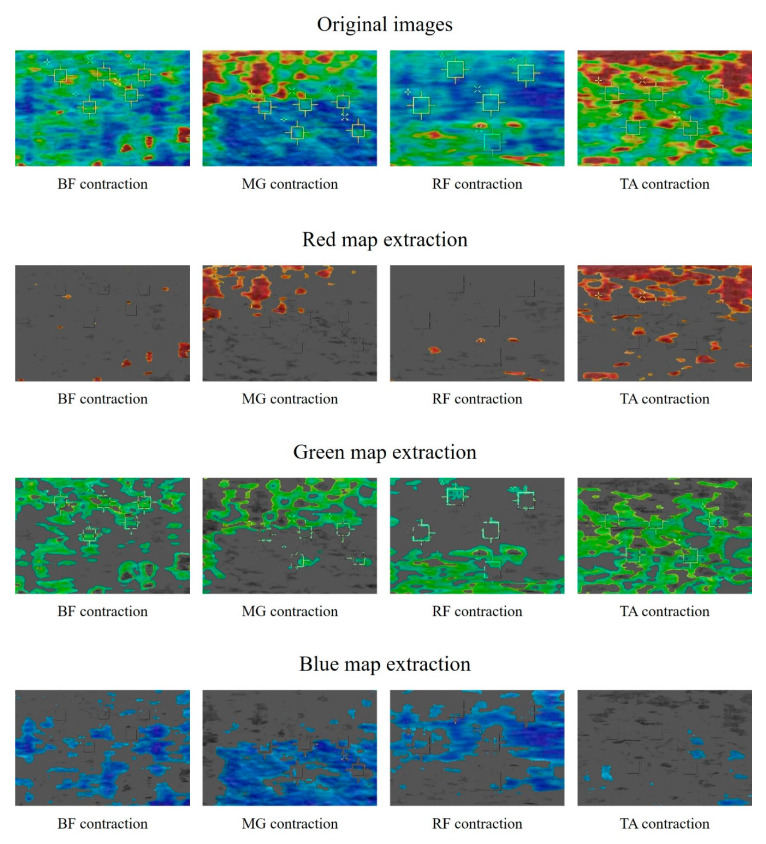
Acquired example images using shear wave elastography for contracted BF, MG, RF, and TA. The images show original red, green, and blue colors and the extraction results for each color.

**Figure 5 diagnostics-10-00362-f005:**
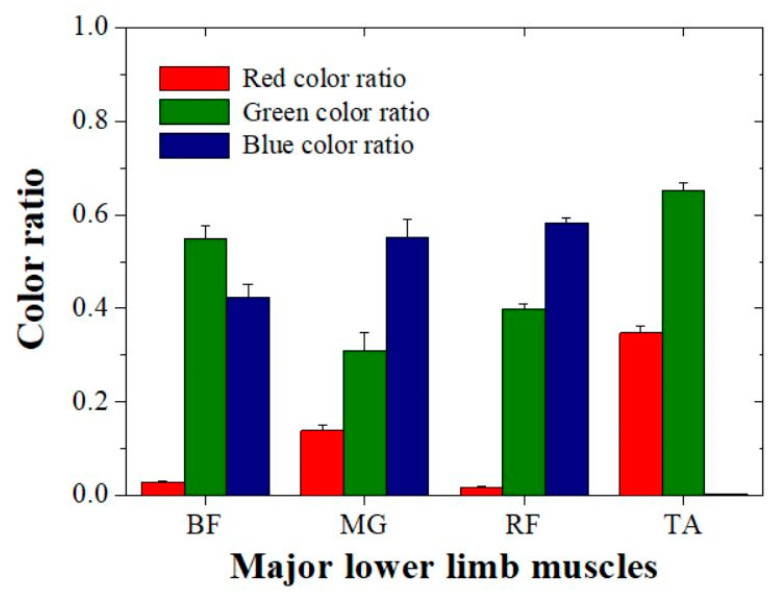
Acquired quantitative color ratio data using our developed program for the major lower limb muscles.

**Table 1 diagnostics-10-00362-t001:** Quantitative evaluation results for color ratio data for red and blue for the major lower limb muscles.

Muscle	Color Ratio Data	
	Red Color	Blue Color
BF	0.028 ± 0.0023	0.423 ± 0.0265
MG	0.139 ± 0.0128	0.552 ± 0.0368
RF	0.018 ± 0.0015	0.583 ± 0.0105
TA	0.347 ± 0.0161	0.002 ± 0.0001

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
