# Peer review of "Development of Stiffness Measurement Program Using Color Mapping in Shear Wave Elastography"

_diagnostics, 2020, doi:10.3390/diagnostics10060362_

Round 1
Reviewer 1 Report
In the manuscript "Development of stiffness measurement program using color mapping in shear wave elastography", Haneul Lee and colleagues performed a study in order to better investigate qualitative shear wave elastography for tissue stiffness in major lower limbs muscles, to improve reliability of shear wave elastography ultrasound imaging. Even though with several limitations, the work has certainly a clinical significance. However, some minor criticisms are present, as follows:
- The authors present the data clear, but in order to better understand the data presented in lines 144-153 , it should be presented in tables.
- Conclusions should be better described in order to highlight the significance and innovation of the work.
Author Response
Dear reviewer,
Thank you for your useful comments and suggestions concerning our paper entitled “Development of stiffness measurement program using color mapping in shear wave elastography”.
We attached response file. Thank you.

Reviewer 2 Report
The authors present a method to quantify shear wave elastography maps without being affected by inter user variability. They utilize the colormap presented by the ultrasound scanner and separate the red, blue, green parts of the colormap to make quantitative assessments. I had the following comments
Major:
Instead of utilizing the colormap to separate the pixels in three bins, this could have been done directly by looking at grayscale maps and separating the values in top 33%, middle 33% and lower 33% of the grayscale range. This would be a more direct way of doing it. Traditionally colormaps have only existed to give the human eye an idea of contrast in their data and I would agree this is a novel way of using the colormap that is usually not done. However, justification why this indirect way was utilized is required
if there are other studies that have similarly used the colormap then please mention them so that your approach can be supported. Additionally, a more thorough literature review of how other researchers have quantified the shear wave maps and other strain maps could benefit the manuscript.
Minor:
Line 55, “In addition, this elastography technique is most widely 55 used for accurate stiffness measurements of muscles or tendon tissues” Please provide citations
Line 72-73, Name of probe used, example. 9L4? How many elements in the probe? Fine pitch is vague, please provide the pitch in mm.
Line 79-80, What is 18 to 30 kg.m^2? is it the BMI? ‘Practice time’ is not a typically used term maybe you want to use the term ‘physical exercise’. Or you could give more information such as ‘practice term playing an outdoor sport’
Line 92 equation 1, I think you mean to say 0≤IR(?,?), IG(?,?), IB(?,?)≤255, since x,y are just pixel indices and IR, IG and IB are the actual intensities
Line 94-96 greater than sign makes more sense to me than less than sign
Line 113-114 Describe the process used to retrieve in MATLAB the elastography maps produced on the Siemens machine by providing more information on if they were copied as DICOM images that were then read in MATLAB or some other process was used.
Line 126 Please describe the operators process to make the ROI
Author Response

(The authors gave the same response as above.)

Round 2
Reviewer 2 Report
I did mention this in the last review but equation 2 (line 104) 'less than' sign might be actually 'greater than' sign. You can change this in post acceptance editing if that is the case.